

# Methylase-assisted subcloning for high throughput BioBrick assembly

Ichiro Matsumura

Emory University School of Medicine, Department of Biochemistry, Atlanta, GA, United States of America

## ABSTRACT

The BioBrick standard makes possible iterated pairwise assembly of cloned parts without any depletion of unique restriction sites. Every part that conforms to the standard is compatible with every other part, thereby fostering a worldwide user community. The assembly methods, however, are labor intensive or inefficient compared to some newer ones so the standard may be falling out of favor. An easier way to assemble BioBricks is described herein. Plasmids encoding BioBrick parts are purified from *Escherichia coli* cells that express a foreign site-specific DNA methyltransferase, so that each is subsequently protected in vitro from the activity of a particular restriction endonuclease. Each plasmid is double-digested and all resulting restriction fragments are ligated together without gel purification. The ligation products are subsequently double-digested with another pair of restriction endonucleases so only the desired insert-recipient vector construct retains the capacity to transform *E. coli*. This 4R/2M BioBrick assembly protocol is more efficient and accurate than established workflows including 3A assembly. It is also much easier than gel purification to miniaturize, automate and perform more assembly reactions in parallel. As such, it should streamline DNA assembly for the existing community of BioBrick users, and possibly encourage others to join.

## INTRODUCTION

A bottleneck in many synthetic biology projects is the physical linkage of cloned synthetic genes ("parts") to each other to form longer functional assemblies ("devices"). The costs of gene synthesis, cloning and DNA sequencing have decreased significantly but syntheses are still limited in length ($\leq$3 kb), nucleotide composition, accuracy and yield (*Kosuri & Church, 2014*; *Kuhn et al., 2017*). Many DNA assembly methods have been invented (*Casini et al., 2015*; *Chao, Yuan & Zhao, 2015*; *Sands & Brent, 2016*; *Vazquez-Vilar, Orzaez & Patron, 2018*; *Watson & Garcia-Nafria, 2019*), which suggests that none work well for every user. The challenges of assembling cloned parts are not identical to those of ligating PCR products into plasmids (*Bryksin & Matsumura, 2010*) so different solutions are demanded.

Many synthetic biologists have adopted cloning standards that stipulate particular type II or type IIS restriction sites at the ends of each DNA "part". The BioBrick RCF[10] standard (*Knight, 2003*) is most established (Fig. 1). All BioBrick-compliant plasmids

Corresponding author
Ichiro Matsumura,
imatsum@emory.edu

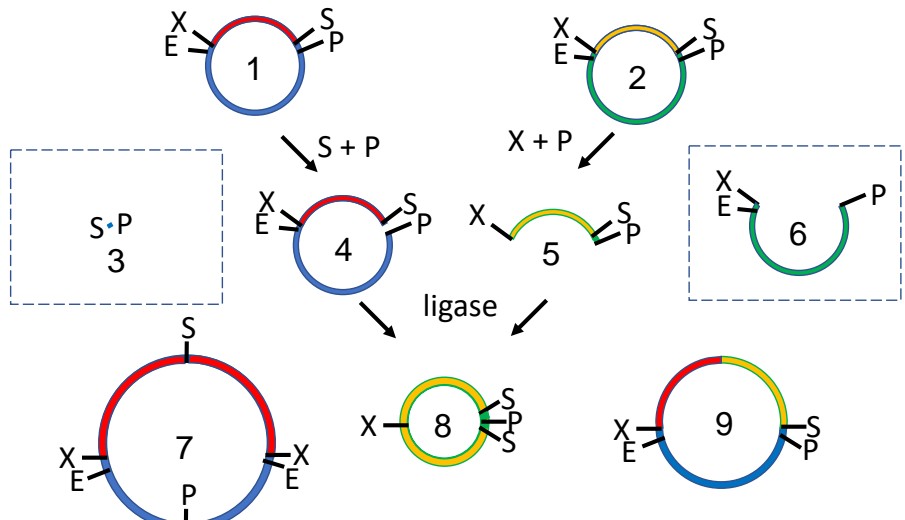

**Figure 1 Conventional subcloning of BioBrick-compatible parts.** Recipient (1) and donor (2) plasmids both contain inserts bound by the same restriction sites (E = EcoRI, X = XbaI, S = SpeI, P = PstI). The recipient plasmid (1) is cut with SpeI and PstI, releasing a short stuffer fragment (or "snippet," 3); the donor (2) is separately cut with XbaI and PstI, so that insert (5) is released from plasmid fragment (6). The fragments from both digests (3-6) are separated by agarose gel electrophoresis. The desired recipient fragment (4) and insert (5) are excised from the gel and subsequently purified; the unwanted stuffer (3) and donor plasmid fragment (6) are left in the gels, which are thrown away. The recipient fragment (4) and insert (5) are ligated together forming three products: the recipient plasmid homodimer (7), the insert homodimer (8) and desired insert-recipient plasmid heterodimer (9). Large inverted repeats (7-8) cannot replicate stably in vivo so the desired construct (9) is the only product capable of conferring antibiotic selection if the digests and ligations were efficient.

contain a characteristic pattern of sites recognized by type II restriction endonucleases (EcoRI-NotI-XbaI-insert-SpeI-NotI-PstI). Two such inserts can be combined by digesting one plasmid (recipient) with SpeI and PstI, and the other (donor) with XbaI and PstI. Alternatively, one plasmid (recipient) can be cut with EcoRI and XbaI, and the other with EcoRI and SpeI (donor). The overhangs of XbaI and SpeI digests products are compatible but anneal to form a "scar" not recognized by either restriction endonuclease. The ligation of the desired insert to the desired recipient plasmid thus creates a new BioBrick-compatible plasmid. The virtue of this approach compared to *ad hoc* subcloning strategies is that an infinite number of inserts can be combined, two at a time, without running out of unique restriction sites. The problem, and focus of this study, is that conventional subcloning (*Matsumura, 2015*), particularly the gel purification step, remains labor-intensive and recalcitrant to automation.

Golden Gate assembly (*Engler et al., 2009*) was invented in part to circumvent gel purifications, though not without some cost. Type IIS restriction endonucleases recognize asymmetric sequences but cut outside of them. BsaI, for example, recognizes the sequence GGTCTC and introduces staggered cuts in both strands downstream regardless of sequence, creating 5′ overhangs that are four nucleotides long. This capacity to create up to 256 different sticky ends with a single enzyme enables concurrent restriction digests and

ligations in a single pot. Such simultaneous reactions will hereafter be called "continuous" to distinguish them from "discontinuous" sequential digestions and ligations. Unlike BioBrick assembly, the Golden Gate method can be used to combine multiple parts in a single reaction. It does not leave the characteristic XbaI/SpeI scar of BioBrick assembly so it is better suited for the fusion of open reading frames.

Golden Gate assembly is not, however, without drawbacks. Any BioBrick part can be adjoined to any other part using standard protocols, including those described here. In contrast, the sticky ends produced by BsaI and other Type IIS restriction enzymes are only compatible with others designed to be complementary. Cloning standards for Type IIS restriction endonucleases, such as MoClo (*Weber et al., 2011*), Phytobricks (*Patron et al., 2015*), Golden Braid (*Sarrion-Perdigones et al., 2011*) or Loop assembly (*Pollak et al., 2019*), facilitate some repurposing of parts for other devices. The MoClo standard, for example, employs nearly three dozen intermediate vectors, each with a unique pair of restriction sites and overhangs, each dedicated to a separate category of parts (e.g., promoters, 5' upstream untranslated regions, open reading frames, terminators etc.) (*Weber et al., 2011*). The BioBrick standard employs a single type of vector (*Knight, 2003*) and a single overhang, created by Type II restriction enzymes XbaI or SpeI, to connect parts. BioBrick assembly experiments are thus relatively easy to plan.

I value the simplicity and universal part compatibility of BioBricks, so I invented a less labor intensive and automation-friendly way to assemble them. The concept that underlies my approach is straightforward and easy to implement. In nature every restriction endonuclease is paired with a corresponding site specific DNA modifying enzyme, most often a methyltransferase (*Loenen & Raleigh, 2014*). Previous reports have described the use of methyltransferases (*Lin & O'Callaghan, 2018*) or methylated primers (*Chen et al., 2013*) to enable Golden Gate assemblies that would otherwise have been impossible. The 2ab assembly method is most relevant to the current study. It utilizes in vivo plasmid methylation and recombination of selectable markers to effect one pot, discontinuous ligations of BglBrick parts using Type II restriction enzymes BglII and BamHI (*Leguia et al., 2013*). It is efficient, requires little labor and amenable to automation. Unfortunately, the BglBrick and BioBrick standards are incompatible. Moreover 2ab assembly requires specialized plasmids encoding pairs of selectable markers. It is nevertheless an important precedent for easier ways to combine BioBrick parts, preferably in existing plasmids.

Here I describe the cloning of relevant methylases and their expression in a laboratory *E. coli* strain. Cells co-transformed with BioBrick-compatible plasmids thus add methyl groups to DNA at specific sites (Fig. 2). The methylated plasmids are prepared and double digested in accordance with traditional cloning protocols, except that smaller quantities of DNA are required. The restriction fragments are not gel purified but rather combined and reacted with T4 DNA ligase. The undesired ligation products, including the original parental plasmids, are subsequently cut by another pair of restriction enzymes. The desired ligation product (insert-recipient plasmid) is protected from both restriction enzymes, so it alone retains the capacity to transform *E. coli*.

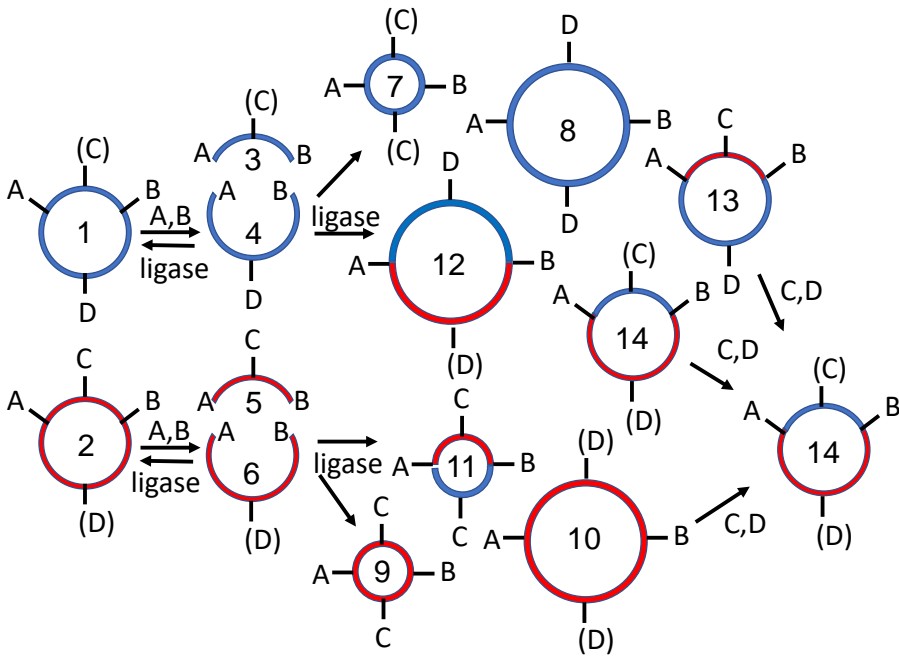

**Figure 2** **Subcloning of a methylated insert into a methylated recipient plasmid.** Donor (1) or recipient (2) plasmids are purified from *Escherichia coli* strains that express foreign DNA methyltransferases that protect restriction sites C or D, respectively. Modified sites are shown in parentheses. Both parental plasmids (1-2) are reacted with restriction enzymes A and B, thereby producing four fragments: insert (3, methylated), donor plasmid fragment (4, unmethylated), stuffer (5, unmethylated), and recipient plasmid fragment (6, methylated). All restriction fragments (3-6) are ligated. Two products recapitulate the parental plasmids (1-2). Four are homodimers (fragments ligating to other copies of themselves, 7-10). The insert (7) and stuffer (9) homodimers resist further digestion but lack any selectable marker or origin of replication. The recipient plasmid (10) and donor plasmid (8) homodimer are also circular, but are both large inverted repeats, so neither is stable in *E. coli*. Four others are heterodimers (11-14). Polymeric concatemers (linear trimers, circular tetramers, etc.) also form at low frequency but are not shown. Double digestion of the ligation products with restriction enzymes C and D linearizes almost all circular products (1-2, 8-9, 11-13) except for the desired double methylated insert-recipient plasmid construct (14). It alone retains the capacity to transform *E. coli* efficiently.

## MATERIAL AND METHODS

### Materials

The synthetic methylase genes used in this study (M.EcoRI, M.XbaI, M.Ocy1ORF8430P, M.PstI, and M.AvaIII) were purchased from IDT (Coralville, IA) as gBlocks. Seakem LE agarose was from Lonza Rockland (Rockland, ME). The 10 bp ladder was from (Thermo Fisher). The other molecular size markers (lambda HindIII, 100 bp ladder), restriction enzymes, T4 DNA ligase and pure bacteriophage lambda DNA were from NEB (Ipswich, MA). TempliPhi rolling circle amplification kits were from Cytiva (Marlborough, MA). MinElute PCR purification and GeneRead Size Selection kits were from Qiagen (Valencia, CA), as was the QIAcube and the custom protocol (vide infra). *E. coli* OmniMax2 cells were from Invitrogen. Ethylenediaminetetraacetic acid (EDTA), L-arabinose and L-rhamnose were from Sigma Chemicals (St. Louis, MO); isopropyl $\beta$-D-1-thiogalactopyranoside
(IPTG) was from Gold Biotechnology (St. Louis, MO). LB broth (Miller) was from EMD Millipore (Billerica, MA) and Bacto-agar was from BD Difco (Franklin Lakes, NJ).

## Methods

### Subcloning via gel purification

Two plasmids were purified in triplicate (from cultures seeded with different colonies) via the Qiagen QIAprep spin miniprep kit. Recipient tagRFP-pUC (1 μg) was digested in 1x NEB CutSmart buffer (80 μL total reaction volume) by EcoRI-HF and XbaI (20 units each), thus releasing a short 15 base pair stuffer fragment ("snippet"); lacI-Ptac-lacO-pUC was similarly digested with EcoRI-HF and SpeI-HF in the same buffer, thereby releasing the lacI-Ptac-lacO insert and pUC donor plasmid. All restriction digests in this study were incubated overnight at 37 °C unless otherwise stated. The desired fragments were separated from the undesired ones in 0.8% LE agarose gels; the bands corresponding to the recipient plasmid tagRFP-pUC and insert tagRFP were excised with a razor blade. The desired DNA was purified from the agarose slices via the QiaQuick gel extraction protocol. The fragments (20 fmol ∼50 ng tagRFP-pUC or 25 ng lacI-Ptac-lacO), alone or in combination, were reacted to T4 DNA ligase (3 Weiss units) in 1x NEB buffer containing 1 mM ATP (20 μL total reaction volume) overnight in temperature cycled reactions (30 °C × 30 s, 10 °C × 30 s) (*Lund, Duch & Pedersen, 1996*). The ligase was heat killed (10 min at 65 °C), and the reactions (1 ng) were used to transform chemically competent OmniMax 2 cells (20 μL). All experiments employed the same batch of cells made competent by the classical method of *Inoue, Nojima & Okayama (1990)*. Transformation efficiency was $3 \times 10^7/\mu g$, as determined by counting colonies after transformation with 10 pg of pUC19.

### Tip Snip subcloning

The lacI-Ptac-lacO-pUC donor plasmid (1μg) in 1× NEB CutSmart buffer (80 μL total reaction volume) was shortened slightly by an extra restriction enzyme (20 units PstI-HF) that recognizes a site adjacent to those used to release the insert (20 units each of EcoRI-HF and SpeI-HF) (*Matsumura, 2017*). The tagRFP-pUC recipient plasmid (1μg) was cut as usual (20 units each of EcoRI-HF and XbaI in 1× NEB CutSmart buffer, 80 μL total reaction volume). The small restriction fragments ("snippets") in both digests are denatured, annealed to exogenously added anti-snippet oligonucleotides (100 nM BioBrick suffix in the donor digestion, 100 nM BioBrick prefix in the recipient digestion), thereby inactivating their sticky ends, and eliminated via Qiagen GeneRead size selection silica spin column chromatography. The purified restriction fragments were ligated (20 fmol ∼60 ng tagRFP-pUC, 90 ng lacI-Ptac-lacO + pUC, 50 nM PstI "unlinker") in temperature cycled NEB T4 DNA ligase buffer (20 μL total reaction volume) prior to heat killing and transformation of *E. coli* as described above.

### 3A assembly

A BioBrick-compatible plasmid that encodes chloramphenicol acetyltransferase, RP4 oriT-pUC57-mini-cat (2 μg) in 1× NEB CutSmart buffer (80 μL total reaction volume) by 20 units each of EcoRI-HF, PstI-HF and NotI-HF (so as to eliminate the sticky ends of its stuffer fragment), dephosphorylated in reactions with NEB Calf Intestinal Phosphatase.

The lacI-Ptac-lacO-pUC donor plasmid (300 ng) in 1x NEB 2. 1 buffer (15 μL total reaction volume) was digested with 6 units each of EcoRI-HF and SpeI; the tagRFP-pUC donor plasmid was similarly digested with XbaI and PstI. The digests containing pUC-mini-cat recipient vector (60 ng), the lacI-Ptac-lacO and tagRFP-pUC inserts (50 ng each) were reacted in a thermocycler with 3 Weiss units of T4 DNA ligase in 1x NEB T4 DNA ligase buffer (10 μL total reaction volume).

### Construction of DNA methyltransferase expression vectors

The methylase expression vectors (Prham-M.EcoRI-p15A-aadA, Prham-M.XbaI-p15A-aadA, Prham-M.Ocy1-p15A-aadA, Prham-M.PstI-p15A-aadA, and Prham-M.AvaIII-p15A-aadA) were constructed as follows. BioBrick compatible DNA methyltransferase genes were synthesized without internal BioBrick restriction sites (EcoRI, NotI, XbaI, SpeI or PstI), cloned into IMBB2.4-pUC57-mini using restriction enzymes EcoRI and PstI, and sequenced. The p15A plasmid origin and spectinomycin resistance marker (aadA) were subcloned from pACYC Duet and pCDF Duet (EMD Millipore, Novagen) respectively into a BioBrick compatible plasmid. The intergenic region between rhaS and rhaB, which includes promoters and operators for both genes, was previously described (*Matsumura, 2017*).

The p15A, aadA, Prham and methylase genes were assembled by a combination of traditional and Tip Snip BioBrick assembly. Leaky expression of M.XbaI or M.Ocy1ORF8430P from BioBricks containing these parts prevented efficient digests of the plasmids with XbaI or SpeI-HF. Those plasmids were amplified in vitro by utilizing the TempliPhi rolling circle protocol. The resulting unmethylated amplification product was subsequently digested, and the desired part was gel purified and ligated to other parts. The BioBrick restriction enzymes (EcoRI, XbaI, SpeI and PstI) were eliminated by digesting the plasmids (or amplified versions of them) with XbaI and SpeI-HF, self-ligating the p15A-aadA-Prahm-methylase and using ligation reaction products to transform *E. coli* OmniMax2. Prham-M.XbaI-p15A-aadA (RRID:Addgene_149338), Prham-M.Ocy1ORF8430P-p15A-aadA (149339), Prham-M.EcoRI-p15A-aadA (149341), Prham-M.AvaIII-p15A-aadA (149342) and Prham-tagRFP-pUC (149343), which was used to optimize the optimal concentration of glucose for auto-induction, have been deposited in the Addgene repository.

### 2RM assembly

Methylated, uncut lacI-Ptac-lacO-pUC and tagRFP-pUC plasmids (240 ng each) were reacted with XbaI, SpeI (6 units each) and T4 DNA ligase (3 Weiss units) in 1x NEB CutSmart buffer supplemented with 1 mM ATP (25 μL total reaction volume) in a single pot reaction analogous to that of Golden Gate assembly (72 cycles of 5 min. at 37 °C, followed by a nested 10 cycles of 30 s at 10 °C and 30 s at 30 °C). The reaction was incubated for another hour at 37 °C, then heat killed for 10 min at 65 °C; 1 ng of total DNA was used to transform 20 μL competent *E. coli* OmniMax 2 cells.

### 4R/2M (PstI)

M.EcoRI-protected lacI-Ptac-lacO-pUC (500 ng) was digested overnight at 37 °C by 6 units of SpeI and 8 units of PstI in 1× NEB 2.1 buffer (25 μL total reaction volume). M.Ocy1-protected tagRFP-pUC was similarly digested by 8 units of XbaI and 12 units of PstI. Note that PstI-HF cannot be heat-killed, nor is SpeI-HF fully active in NEB 2.1 buffer, so PstI and SpeI were utilized instead. The restriction enzymes were heat-killed (20 min at 80 °C), and the restriction fragments (45 ng tagRFP-pUC, 10 ng tagRFP + pUC) were reacted to T4 DNA ligase (2.4 Weiss units in 1x NEB 2.1 buffer supplemented with 1 mM ATP, 20 μL total reaction volume) overnight in a thermocycler (600 cycles of 30 s at 30 °C, 30 s at 10 °C). The ligase was heat killed by incubation at 65 °C for 10 min. A 2 μL aliquot of each ligation was diluted into a 26 μL 1x NEB 2.1 buffer containing 8 units each of EcoRI-HF and SpeI. The post-ligation digest was incubated for 3 h at 37 °C, and 1 μL of the reaction was used to transform 20 μL of competent *E. coli* OmniMax 2 cells.

## RESULTS

### Subcloning via gel purification as a gold standard

Established subcloning methods (*Matsumura, 2015*) were initially applied to set quantitative benchmarks for efficiency (number of correctly assembled clones per ng ligated DNA) and accuracy (fraction of correctly assembled clones among total). Efficiency is important because it is an indirect measure of reliability when optimal conditions cannot be achieved. Two plasmids, lacI-Ptac-lacO-IMBB2.4-pUC57-mini and tagRFP-IMBB2.4-pUC57-mini (hereafter abbreviated lacI-Ptac-lacO-pUC and tagRFP-pUC respectively) were selected as models for this study (Fig. 3). Both comply with requirements for established BioBrick RFC[10] assembly protocols. Colonies of cells transformed with the desired assembly product, lacI-Ptac-lacO-tagRFP-pUC, turn pink due to leaky expression of the fluorescent marker protein. Throughout this study, the same *E. coli* strains, DNA purification techniques, restriction enzymes, ligases and reaction buffers were used, generally in accordance with manufacturer's instructions except as noted. Differences in outcome can thus be attributed solely to differences in assembly protocols. Each cloning step was carried out in triplicate, starting with individual isolated bacterial colonies; standard errors are reported as a measure of variation between experimental replicates.

The most labor-intensive steps of a traditional subcloning experiment are the separation of restriction fragments via agarose gel electrophoresis, excision of bands corresponding the desired fragments and the extraction of DNA from the agarose slice. Overnight incubations of transformed bacteria, restriction digests and temperature cycled ligation reactions were rate-limiting. The aim here was not to accelerate the workflow, but rather to decrease labor input and increase throughput without compromising efficiency or accuracy. After restriction digests, gel purification and ligation, transformation of *E. coli* with the ligation products led to the growth of $126 \pm 44$ pink colonies per ng; a minority of white colonies ($11 \pm 4 = 8\%$) grew on those LB-ampicillin plates (Table 1). The background on control plates spread with cells transformed with vector only ligations was low ($7 \pm 2$ cfu/ng), which suggested that restriction digests were nearly complete. The insert only ligation

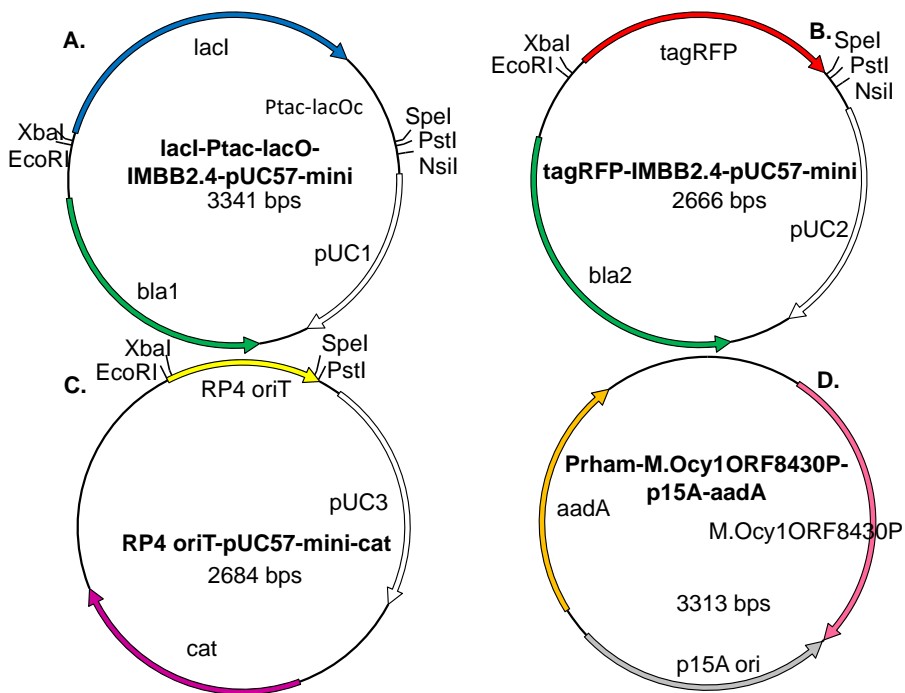

**Figure 3  Model plasmids used in this study.** The lacI-Ptac-lacO insert (A) includes a promoter that is somewhat leaky at high copy number. The IMBB2.4-pUC57-mini backbone (A–B), hereafter abbreviated pUC, is BioBrick-compatible and also includes an NsiI site downstream of PstI (*Matsumura, 2017*). The tagRFP reporter (B) protein can cause colonies to turn visibly pink, but only when the gene encoding it is subcloned downstream of a leaky or constitutive promoter. RP4 oriT-pUC-cat (C) is a BioBrick compatible plasmid that confers resistance to chloramphenicol instead of ampicillin. RP4 oriT serves as a small stuffer in these experiments. In this study this latter plasmid is used only as a recipient plasmid (destination vector) for 3A assembly. Five expression vectors for production of recombinant DNA methyltransferases were constructed for this study. The version that expresses M.Ocy1ORF8430P, a putative ortholog of M.SpeI, is shown (D). The others are similar in design but express M.XbaI, M.EcoRI, M.PstI or M.AvaIII instead. Each plasmid utilizes the low copy number p15A origin (pACYC) and confers resistance to spectinomycin and is thus compatible with pUC plasmids that impart resistance to ampicillin, chloramphenicol or kanamycin. The DNA methyltransferase expression vectors do not contain any of the restriction sites employed in BioBrick assembly protocols (EcoRI, XbaI, SpeI or PstI), so they will not produce restriction fragments that ligate to those that are desired.

controls produced greater background (61 ± 27 cfu/ng), which suggests that the insert was not effectively separated from the donor plasmid in this experiment. Two other established subcloning techniques, tip snip (*Matsumura, 2017*) and 3A (*Shetty et al., 2011*), were also used to provide standards of comparison (Table 1, also described in Supplemental Materials).

## Methylase expression vectors

The overarching strategy of this study is to replace the gel purification step of subcloning by a combination of site-specific DNA methylation and post-ligation restriction digestion (*Spear, 2000*; *Zeng, Eidsness & Summers, 1997*). To realize this strategy, BioBrick compliant genes encoding the DNA methyltransferases of the EcoRI, XbaI and PstI restriction
**Table 1  Colony counts (cfu/ng).**

| Assembly protocol | Vector only | Insert only | Vector + insert (red) | Vector + insert (white) |
| --- | --- | --- | --- | --- |
| Gel purify (EcoRI) | $7 \pm 2$ | $61 \pm 27$ | $126 \pm 44$ | $11 \pm 4$ |
| Tip Snip (EcoRI) | $8 \pm 2$ | $9 \pm 6$ | $384 \pm 61$ | $11 \pm 4$ |
| 3A | 0 | 0 | $4 \pm 1$ | $5 \pm 2$ |
| 2RM | $20 \pm 4$ | $4 \pm 1$ | $118 \pm 13$ | $260 \pm 25$ |
| 4R/2M (PstI) | $1 \pm 0.2$ | $1 \pm 0.2$ | $177 \pm 4$ | $2 \pm 1$ |
| 4R/2M (EcoRI) | $0 \pm 0$ | $2 \pm 1$ | $19 \pm 7$ | $8 \pm 6$ |
| 4R/2M (EcoRI,NsiI) | $1 \pm 1$ | $10 \pm 4$ | $299 \pm 91$ | $12 \pm 3$ |

modification systems were synthesized, cloned into compatible plasmids and sequenced. The complete sequence of SpeI methylase (M.SpeI) is not available on REbase (*Roberts et al., 2010*), so a putative ortholog M.Ocy1ORF8430P (hereafter abbreviated M.Ocy1) was synthesized instead. Each DNA methyltransferase gene was subcloned via traditional techniques downstream of the T5 (*Bujard et al., 1987*), tac (*De Boer, Comstock & Vasser, 1983*) and rhamnose operon (*Egan & Schleif, 1993*) promoters and a strong ribosome binding site.

The promoter-methylase expression cassettes were subcloned into a simple plasmid consisting only of the p15A replication origin, which is low in copy number and compatible with more common plasmids that encode the pUC origin, and streptomycin $3''$-adenylyltransferase (aadA) selectable marker (Fig. 3). The new expression plasmids (promoter-methylase-p15A-aadA) confer resistance to streptomycin and spectinomycin. They don't contain any of the restriction sites normally used for BioBrick assembly (e.g., EcoRI, XbaI, SpeI or PstI) so they won't release any restriction fragments that would interfere with any downstream subcloning steps.

The in vivo methylase activities produced by these expression vectors was tested as follows. *E. coli* strain OmniMax 2 was co-transformed with each vector and another BioBrick compatible plasmid, propagated to mid-log culture and induced (either with IPTG or L-rhamnose) for three hours. The plasmids were purified and reacted with restriction endonucleases including the one normally associated with each DNA methyltransferase in wild-type bacteria. The degree of protection was assessed by comparing the mobilities in agarose gels of plasmids that were uncut, completely cut by a restriction endonuclease unrelated to the methylase or protected at least in part by in vivo methylation. For example, agarose gel electrophoresis showed that lacI-Ptac-lacO-pUC purified from *E. coli* carrying Prham-M.XbaI-p15A-aadA was digested by SpeI but mostly resistant to XbaI. Conversely, tagRFP-pUC protected by Prham-M.Ocy1-p15A-aadA was digested with XbaI but mostly resistant to SpeI (Fig. 4).

The rhamnose promoter, reputedly the weakest of the three tested, proved most reliable for consistent and complete in vivo methylation. I speculate that DNA methyltransferases that are site-specific at moderate concentrations become toxic to host cells when over-expressed (*Bandaru, Gopal & Bhagwat, 1996*). Extended over-expression could thus favor the accumulation of mutations beneficial to transformed cells but unwanted by human scientists. Induction of transformants at mid-log phase is itself labor-intensive, as cultures

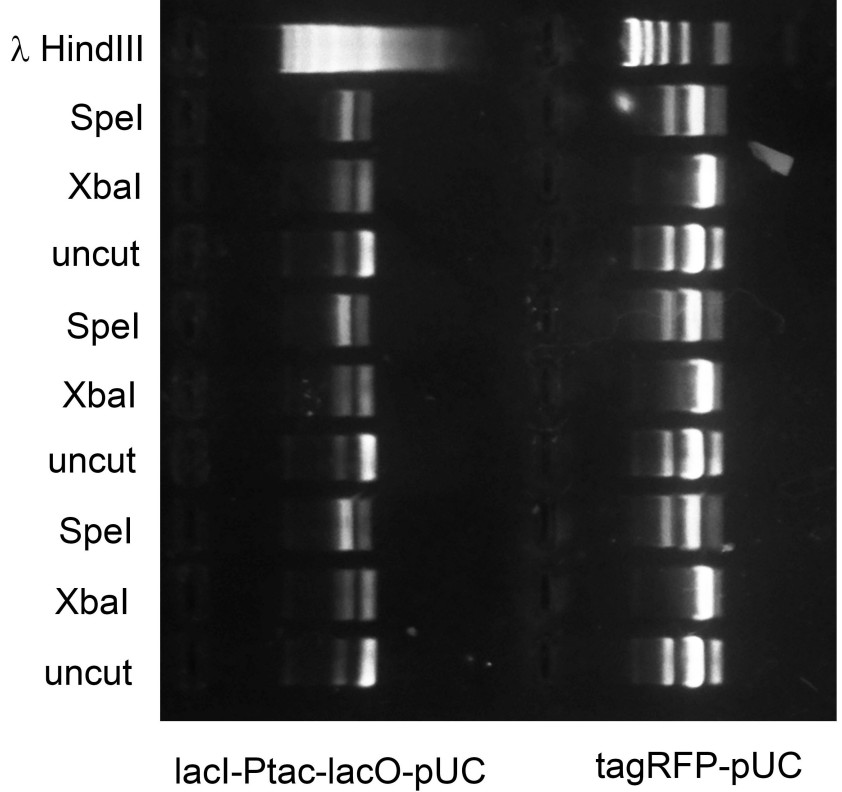

λ HindIII

SpeI

XbaI

uncut

SpeI

XbaI

uncut

SpeI

XbaI

uncut

lacI-Ptac-lacO-pUC          tagRFP-pUC

**Figure 4  M.XbaI and M.Ocy1ORF8430P protect plasmids from XbaI and SpeI.** Model plasmids lacI-Ptac-lacO-pUC and tagRFP-pUC were purified from triplicate cultures of *E. coli* OmniMax 2 co-transformed with Prham-M.XbaI-p15A-aadA or Prham-M.Ocy1ORF8430P-p15A-aadA (Fig. 3D) respectively. Each purified enzyme was reacted in vitro with XbaI or SpeI-HF, and the extent to which each was cut was assessed by agarose gel electrophoresis. Each of the DNA methyltransferases appears to protect cohabiting plasmid from its corresponding restriction endonuclease, and that protection is sequence specific.

propagated in parallel don't always grow at the same rate, so an auto-induction protocol was developed. The rhamnose promoter is regulated by catabolite repression as well as by L-rhamnose. The plasmid Prham-tagRFP-pUC (*Matsumura, 2017*) was used to transform *E. coli* OmniMax 2. Limiting amounts of glucose were added to saturating concentrations of L-rhamnose (0.1%) in LB medium supplemented with ampicillin. Commercial LB contains varying quantities of glucose, but for the addition of 0.001% glucose to 0.1% L-rhamnose led to maximum tagRFP expression as measured in a microtiter plate spectrofluorimeter. Autoinduction under those growth conditions led complete in vivo methylation when the methylase expression vectors were used instead.

The other lesson inferred from the in vivo methylation experiments was that M.PstI is rarely able to methylate plasmids within *E. coli* cells as completely as M.EcoRI, M.XbaI or M.Ocy1. Each of these methylases evolved in a different bacterial species so it isn't surprising that one of the four proved less active than the others in the alien environment of the *E. coli* cytoplasm. Most of our plasmids include an NsiI site adjacent to the PstI

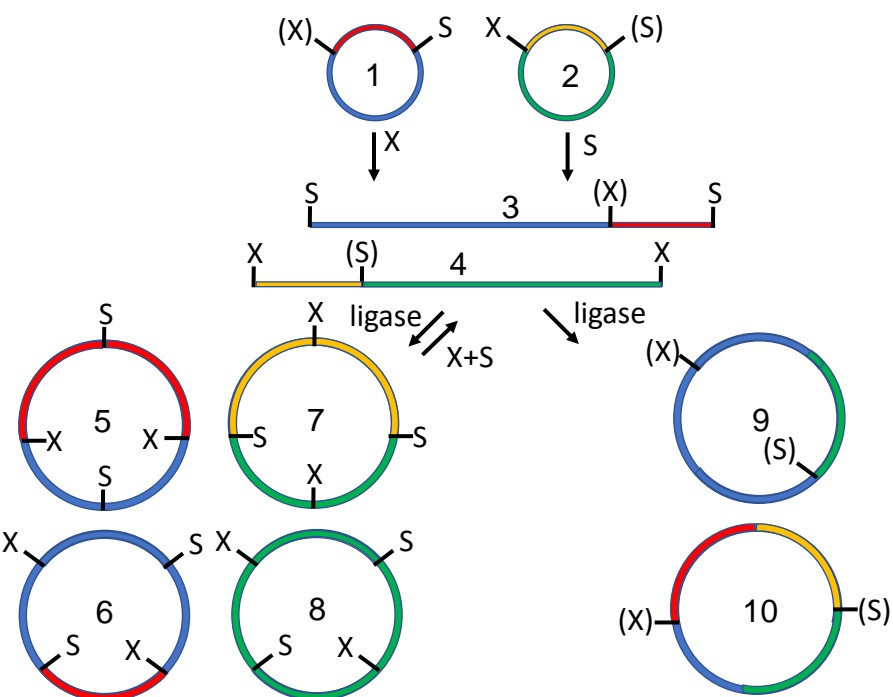

**Figure 5** **2RM BioBrick assembly.** The XbaI site of one cloned BioBrick part (1), and the SpeI site of another (2), are methylated in vivo. These methylated plasmids are mixed together with XbaI, SpeI and T4 DNA ligase. Each plasmid is digested by one restriction endonuclease and protected from the other. The linear digestion products (3-4) are self-ligated to form the parental plasmids (1-2), to another copy of the same molecule in one of two orientations to form a homodimer product (5-8) or to a copy of the other plasmid (again in one of two possible orientations) to form a heterodimer (9-10). The parental plasmids (1-2) and homodimers (5-8) are susceptible to re-digestion, so they are depleted over time while the heterodimers (9-10) accumulate. Both the desired (10) and undesired (9) heterodimers replicate in vivo so restriction mapping is required to differentiate between the two.

site. The sequence of M.NsiI was not available on REbase (*Roberts et al., 2010*) so the gene encoding the M.AvaIII ortholog was synthesized, cloned, sequenced and subcloned downstream of the rhamnose promoter. M.AvaIII proved much more adept at methylating plasmids in the *E. coli* cytoplasm than did M.PstI.

## 2RM assembly

The potential utility of the methylase expression vectors was demonstrated in a series of assembly experiments. The 3A BioBrick assembly protocol (*Shetty et al., 2011*) was so named because it employs three plasmids, each with a distinct antibiotic selection marker. For similar reasons, 2RM assembly utilizes the components of two restriction modification systems: restriction endonucleases XbaI and SpeI-HF, and DNA methyltransferases M.XbaI and M.SpeI homologue M.Ocy1 (Fig. 5). In this embodiment, the lacI-Ptac-lacO-pUC was purified from triplicate cultures of auto-induced cells containing Prham-M.XbaI-p15A-aadA, while tagRFP-pUC was purified from cultures co-transformed with Prham-M.Ocy1-p15A-aadA. The purified plasmids were mixed and reacted with XbaI and SpeI. Each plasmid, lacI-Ptac-lacO-pUC and tagRFP-pUC, was cut with one of the two restriction

enzymes and protected by methylation from the other. The linearized plasmids (Fig. 5 and Fig. S1) react with T4 DNA ligase to form three sets of products. Most common, presumably, are the two original parental plasmids. Each of the linearized plasmids can also be ligated to other copies of themselves in one of two orientations to form homodimers (Fig. 5 and Fig. S2). All contain unmethylated XbaI or SpeI sites, so they are susceptible to re-digestion by the restriction enymes in the reaction vessel. The linearized plasmids can also ligation to each other to form heterodimers (Fig. 5 and Fig. S3). These products are resistant to both restriction endonucleases so they should accumulate over the course of the digestion/ligation reaction.

When *E. coli* were transformed with one nanogram of each ligation reaction, $118 \pm 13$ pink cfu/ng and $260 \pm 25$ white cfu/ng were observed on each plate (Table 1). Colony numbers on plates corresponding to control ligations with only one plasmid ($20 \pm 4$ cfu/ng) or the other ($4 \pm 1$ cfu/ng) were relatively low, suggesting that both methylation and restriction digestion was nearly complete. These results in combination show that restriction digestion of the parental plasmids and homodimeric ligation products was efficient, and that ligation to form heterodimeric products was also efficient. In principle, the ratio of pink to white colonies should be 1:1, but the 1:2.2 ratio observed here could mean that the ligation product with the undesired orientation conferred greater fitness upon the host cell. The desired product contains two copies of the selectable marker and origin of replication (Fig. 5 and Fig. S3), which could complicate subsequent assembly reactions. Double digests of existing BioBrick-compatible plasmids enable directional cloning, which is more practical.

## 4R/2M (PstI) assembly

In 4R/2M assembly, the two parental plasmids are sequentially reacted with two DNA methyltransferases, three restriction endonucleases, T4 DNA ligase and a fourth restriction enzyme (Fig. 6). In its 4R/2M (PstI) embodiment, the recipient encodes the part that will end up on the 5′ end of the desired ligation product. Its EcoRI site is methylated in vivo and subsequently digested by SpeI and PstI (Fig. 6 and Fig. S4). The donor plasmid that encodes the insert destined for the 3′ end of the desired ligation product; it is protected from SpeI by M.Ocy1 and separately double digested by XbaI and PstI. The restriction endonucleases in both digestion reactions are subsequently heat-killed (20 min. at 80 °C); the four digestion products are combined and reacted with T4 DNA ligase and ATP. The ligase is then heat-killed, and the ligation products (Fig. 6, Figs. S5 and S6) are diluted and further digested with EcoRI and SpeI.

EcoRI linearizes the undesired donor plasmid and any ligation product that includes it. SpeI linearizes the other parental plasmid, so that the desired insert-recipient plasmid ligation product is the only viable construct that remains intact. Homodimeric constructs are produced in any ligation of fragments produced by type II restriction endonucleases (Fig. 1), but none are viable in vivo because plasmids are destabilized by large inverted repeats. Competent *E. coli* were transformed with the 4R/2M (PstI) assembly reactions, leading to the formation $177 \pm 4$ pink cfu/ng and only $2 \pm 1$ white cfu/ng (Table 1). Background colony counts on the control plates representing vector only ($1 \pm 0.2$ cfu/ng)

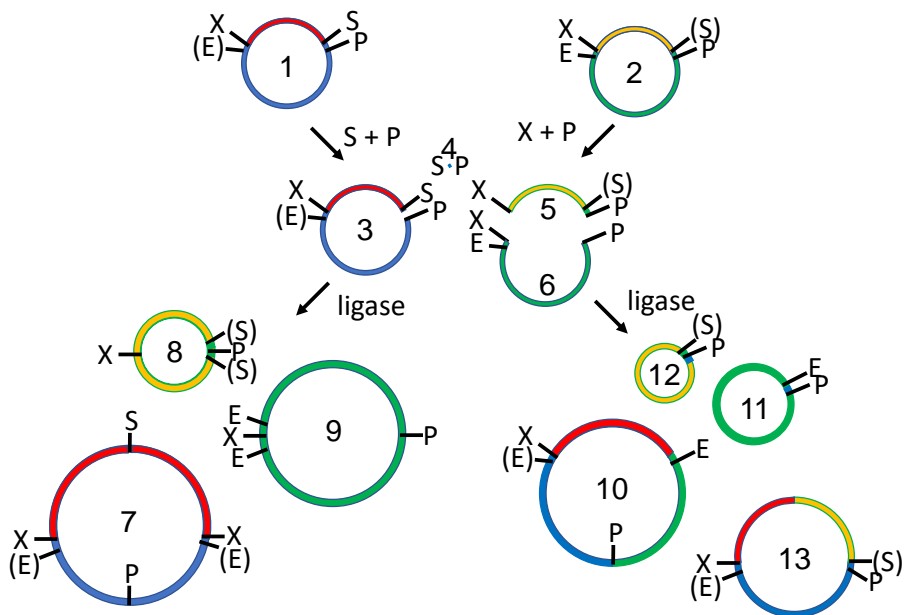

**Figure 6  4R/2M (PstI) BioBrick assembly.** The EcoRI site of the recipient plasmid (1) and SpeI site of the insert (2) are methylated in vivo. The recipient plasmid (1) is digested with SpeI and PstI, so that it releases a short 18 bp stuffer (or "snippet", 4). The donor plasmid (2) is separately digested with XbaI and PstI, producing the desired insert (5) and the undesired donor plasmid fragment (6). The restriction enzymes are heat-killed, the digestion products are mixed and reacted with T4 DNA ligase, forming three sets of ligation products: parental-plasmids (1-2), homodimers (7-9) and heterodimers (10-13). The 36 bp snippet homodimer is not shown, nor are trimer, tetramer and other higher order products. The homodimer products (7-9) are large perfect inverted repeats, which are not expected to replicate efficiently in vivo. Moreover, none of the undesired parental (1-2), homodimer (7-9) or heterodimers (10-11) are resistant to subsequent digestion with EcoRI and SpeI. Only the desired insert/recipient recombinant plasmid (13) retains its ability to transform *E. coli*.

and insert only (1 ± 0.2 cfu/ng) ligations were very low. The 4R/2M (PstI) assembly is thus well suited for routine high throughput BioBrick assembly. I have subsequently used it to assemble 65 more pairs of BioBricks in batches of up to 18.

## 4R/2M (EcoRI) assembly

The logic of 4R/2M (EcoRI) BioBrick assembly is identical to that of 4R/2M (PstI), except that the recipient and donor plasmids are switched. The BioBrick part that ends up on the 5′ end of the assembled product is the insert rather than part of the recipient plasmid. The recipient tagRFP-pUC was methylated in vivo by M.PstI; 600 ng was double digested by EcoRI-HF and XbaI (12 units each in 30 μL NEB CutSmart buffer). Donor lacI-Ptac-lacO-pUC was protected by M.XbaI prior to purification; 600 ng was similarly digested with EcoRI-HF and SpeI-HF (Fig. S7). The restriction enzymes in both digests were heat-killed (20 min. at 80 °C) and the restriction fragments (50 ng tagRFP-pUC, 90 ng lacI-Ptac-lacO) were mixed and reacted overnight in a thermocycler with T4 DNA ligase (3 Weiss units in 25 μL NEB CutSmart buffer supplemented with 1 mM ATP). The enzyme was heat-killed (10 min. at 65 °C), and the ligation product (1 ng/μL) digested with 8 units

each PstI-HF and XbaI in NEB CutSmart buffer (Figs. S8 and S9). The transformation of competent *E. coli* cells produced only 19 ± 7 pink colonies, significantly less than the 4R/2M (PstI) experiment with the same plasmids, and 8 ± 6 white colonies per ng (Table 1). As previously noted, M.PstI does not methylate in vivo as reliably as our other DNA methyltransferases.

The assembly was repeated, except that the tagRFP-pUC plasmid was reacted in vivo with M.AvaIII instead of M.PstI. M.AvaIII catalyzes the methylation of NsiI sites, which exist in most BioBrick compatible plasmids in my lab (*Matsumura, 2017*). NsiI produces sticky ends compatible with those of PstI so it offers a good comparison. This assembly, after digestion with NsiI and XbaI, produced 299 ± 91 pink colonies and only 12 ± 3 white colonies per ng (Table 1). This improved result in consistent with the hypothesis that 4R/2M assembly can be limited by the degree to which the populations of plasmids purified from *E. coli* are methylated.

## DISCUSSION

The assembly protocols described here could be further improved in several ways. The 4R/2M (EcoRI) is more efficient when M.AvaIII expression vectors were employed instead of those that produce M.PstI. Not all BioBrick compatible plasmids contain NsiI sites, so in vivo M.PstI activity could be enhanced, either by optimizing expression via directed evolution (using *in vitro* PstI activity as a selection), co-expression with the PstI restriction endonuclease (as in the wild-type operon) or by identifying an M.PstI ortholog that is more active in the *E. coli* cytoplasm. Another alternative is to clone and express another site-specific DNA methyltransferase that protects some other site that is common in plasmid backbones but very rare within inserts. The tactic of using pairs of methylases to protect desired insert-recipient plasmids from double digests following ligation need not be restricted to BioBrick assembly. It could potentially be generalized to streamline other kinds of subcloning experiments if the relevant DNA methyltransferase expression vectors were available.

The 2RM assembly method is a single pot continuous reaction for the restriction digestion and ligation of BioBrick parts, analogous to Golden Gate assembly except that half or more of the recombinant plasmids are ligated in the undesired orientation. The utility of the existing protocol is limited, but it offers some evidence that continuous assembly of correctly oriented ligation products is possible. Such a process would probably require a more elaborate variant of the BioBrick standard and plasmids methylated at more than one restriction site. If four Type II restriction endonucleases and T4 DNA ligase work together efficiently, two steps (heat killing restriction enzymes, ligation reaction setup) of the 4R/2M protocol would be obviated. This hypothetical assembly process would retain the simplicity of the BioBrick standard but emulate the ease of use of Golden Gate.

## CONCLUSIONS

The 4R/2M (PstI) BioBrick assembly described above is less labor-intensive than is the traditional gel purification approach. It is more efficient and accurate than is 3A assembly

and requires less reagents than does Tip Snip subcloning. The value of the labor savings is proportional to the number of assemblies that can be conducted in parallel. The 4R/2M procedure was not designed to match the convenience of single pot, continuous Golden Gate assembly, but BioBrick assembly experiments are arguably easier to design and debug. The BioBrick standard thus remains well suited for the high school and undergraduate students who participate in iGEM competitions. The throughput of 4R/2M BioBrick assembly is mostly limited by the numbers of plasmid minipreps that users can perform in parallel. The quantity of plasmid required is relatively low ($\leq$400 ng/digest, as opposed to 1–2 $\mu$g for gel purification or Tip Snip) because none is lost during subsequent spin column chromatography. This methodological advance should thus accelerate the work of the BioBricks user community and encourage others to join.

## ACKNOWLEDGEMENTS

Ichiro Matsumura thanks Shanthi Hegde, Srikusumanjali Pinnamareddy (both from Lambert High School, Suwanee, GA), Grace Yan and Emily Yang (both from George Walton Comprehensive High School, Marietta, GA) for beta-testing the 4R/2M (EcoRI) BioBrick assembly.

### Funding

This work was supported by the National Science Foundation (MCB 1359575, MCB 1413062). The funders had no role in study design, data collection and analysis, decision to publish, or preparation of the manuscript.

### Grant Disclosures

The following grant information was disclosed by the author:
National Science Foundation: MCB 1359575, MCB 1413062.

### Competing Interests

The author declares that he has no competing interests.

### Author Contributions

- Ichiro Matsumura conceived and designed the experiments, performed the experiments, analyzed the data, prepared figures and/or tables, authored or reviewed drafts of the paper, and approved the final draft.

### DNA Deposition

The following information was supplied regarding the deposition of DNA sequences:
The DNA methyltransferase expression vectors and their sequences are available from Addgene (IDs: 149338, 149339, 149341, 149342, 149343).

### Data Availability

Raw data are available as a Supplemental File.

## Supplemental Information

Supplemental information for this article can be found online at http://dx.doi.org/10.7717/peerj.9841#supplemental-information.

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
