# Peer review of "Methylase-assisted subcloning for high throughput BioBrick assembly"

_PeerJ, doi:10.7717/peerj.9841_

## Round 0.1 · original submission · Major Revisions

Dear Dr. Matsumura:

Thanks for submitting your manuscript to PeerJ. I have now received two independent reviews of your work, and as you will see, the reviewers raised some minor concerns about the research (mostly the manuscript format and content). Despite this, these reviewers are optimistic about your work and the potential impact it will have on research studying improvements to the BioBrick standard for iterated pairwise assembly of cloned parts without any depletion of unique restriction sites. Thus, I encourage you to revise your manuscript, accordingly, taking into account all of the concerns raised by both reviewers.

While the concerns of the reviewers are relatively minor, this is a major revision to ensure that the original reviewers have a chance to evaluate your responses to their concerns. There are not too many suggestions; thus, it should not take much effort to address these concerns to greatly improve your manuscript.

I look forward to seeing your revision, and thanks again for submitting your work to PeerJ.

Good luck with your revision,

-joe

·

Basic reporting

The quality of the language and writing is excellent. I found the paper clear, and the general subject of assembly chemistries well-reviewed and insightful. The experiments performed are adequate evidence of the claims made. There is a prior study from my own lab: "2ab assembly: a methodology for automatable, high-throughput assembly of standard biological parts" PMID: 23305072 that describes a similar method to the subject of this paper, and is worth comparing. The 2AB method was a BioBrick-like scheme employing the in vivo methylation of BamHI and BglII restriction sites. Those methyltransferases were well behaved and sufficiently complete in their methylation to enable one-pot assembly reactions.

Experimental design

I found this research interesting. The purpose of the research was very clear. Doing DNA chemistry in vivo is a challenging subject, and each new system presents new insights. It is interesting to see this applied to the BioBrick system. It is fine for the journal. The experiments were performed and described appropriately.

Validity of the findings

I suspect that In vivo assembly methods will one day eclipse the in vitro methods. This publication does not achieve that lofty goal, but I found the work insightful and presenting an interesting datapoint about controlling DNA modification reaction in vivo.

I was a little unclear about the claims about one-pot reactions. The statement "The 2RM assembly technique shows how site-specific methylation could potentially enable one pot digestion/ligation reactions." sounds like you are saying that one-pot doesn't work yet. But in other places it seems like you are saying it does work, and I don't see any description of what goes wrong if that does not work. There are protocols talking about stepwise digestions, and others as one-pot. I got a little lost on those details.

Reviewer 2 ·

Basic reporting

This manuscript presents a new method for assembling BioBricks using differential methylation and related non-gel-based DNA ligation approaches. The idea seems sound, the implementation is detailed in general though missing some key analyses. The presentation, however, limits the impact of this work.

Experimental design

No comment.

Validity of the findings

The manuscript presents colony counts as the main evidence to support its claims on correct plasmid assembly. This is indirect evidence and as such I would suggest that a sample of the resulting plasmids are sequenced to directly confirm that the plasmid was assembled as expected and to screen for any systematic sequence artifacts that may arise.

Additional comments

Major presentation issues:
1. Thank you for a thorough description of your technique and thought process. However, your message is lost amidst the figures. In general, they would benefit from being more schematic and displaying only the information necessary to get the point across. Figure 1 does a better job at this by only labeling the relevant sites and color coding the general segments of the plasmids. However, although Figure 1 summarizes the technique successfully, it would be better to organize the method in step-wise columns so the figure can convey the same information quicker. In particular, a large figure side-by-side comparisons of the different cloning techniques utilized in the manuscript would be extremely useful (gold standard, trip snip, 3A, 2RM and 4R/2M).
2. Figures 3,4,7-15 share enough context to consolidate into a fewer, maybe a single large figure that summarizes all the cloning comparisons. Figures 2 and 5 can also be combined.
3. Figure 5 is not mentioned in the main text.
4. Why are none of the students acknowledged authors of the manuscript?
5. Some examples where the language could be improved and or concepts could be expanded upon include:
a. Line 51. Can you expand and provide references on what you mean by improvements in ‘miniturization and automation’?
b. Line 61. The current BioBrick RCF paradigm could use a Figure.
c. Line 71. For the sake of clarity, it would be useful to have a Figure that shows the gold standard process and highlight where the pain point is (i.e. gel purification step).
d. Lines 93 and 99. Can you provide evidence about BioBrick technology being ‘easier to comprehend’ and your method being ‘easier to understand’?
e. Line 102. Can you be more specific with the phrasings ‘protect sites within inserts’ [from what?] and ‘mark sites at their ends’?
f. Line 104. Use a different term than ‘decorate DNA’ to describe the addition of methyl groups to DNA.
g. Line 106. What do you mean by ‘smaller scale’?
h. Line 144. Include how did you determine transformation efficiency in methods.
i. Line 268. Can you provide further evidence about the 3A method not ‘working well’ in people’s hands? Also, can you be more specific about what you mean by ‘working well’?
j. Line 258. Grammar, ‘ligation’ should be ‘ligate’.
k. Line 443. Grammar? What do you mean by ‘multiply methylated’?
6. I recommend using RRIDs (Research Resource Identifiers) to better document the materials used in the experiments.

---

## Round 0.2 · accepted · Accept

Dear Dr. Matsumura:

Thanks for revising your manuscript based on the concerns raised by the reviewers. I now believe that your manuscript is suitable for publication. Congratulations! I look forward to seeing this work in print, and I anticipate it being an important resource for researchers studying improvements to the BioBrick standard for iterated pairwise assembly of cloned parts without any depletion of unique restriction sites. Thanks again for choosing PeerJ to publish such important work.

Best,

-joe

·

Basic reporting

I am fine with the modified manuscript

Experimental design

I am fine with the modified manuscript

Validity of the findings

I am fine with the modified manuscript

Additional comments

I am fine with the modified manuscript